# Cephalosporin translocation across enterobacterial OmpF and OmpC channels, a filter across the outer membrane

Muriel Masi [1,4], Julia Vergalli[1,4], Ishan Ghai[2], Andrea Barba-Bon[2], Thérèse Schembri[1,3], Werner M. Nau [2], Daniel Lafitte[1,3], Mathias Winterhalter [2] & Jean-Marie Pagès [1,4✉]

Gram-negative porins are the main entry for small hydrophilic molecules. We studied translocation of structurally related cephalosporins, ceftazidime (CAZ), cefotaxime (CTX) and cefepime (FEP). CAZ is highly active on *E. coli* producing OmpF (Outer membrane protein F) but less efficient on cells expressing OmpC (Outer membrane protein C), whereas FEP and CTX kill bacteria regardless of the porin expressed. This matches with the different capacity of CAZ and FEP to accumulate into bacterial cells as quantified by LC-MS/MS (Liquid Chromatography Tandem Mass Spectrometry). Furthermore, porin reconstitution into planar lipid bilayer and zero current assays suggest permeation of ≈1,000 molecules of CAZ per sec and per channel through OmpF versus ≈500 through OmpC. Here, the instant killing is directly correlated to internal drug concentration. We propose that the net negative charge of CAZ represents a key advantage for permeation through OmpF porins that are less cation-selective than OmpC. These data could explain the decreased susceptibility to some cephalosporins of enterobacteria that exclusively express OmpC porins.

[1] MCT, UMR_MD1, U-1261, Aix-Marseille Université, INSERM, IRBA, Faculté de Pharmacie, 27 Boulevard Jean Moulin, 13005 Marseille, France. [2] Department of Life Sciences and Chemistry, Jacobs University Bremen, 28719 Bremen, Germany. [3] MCT, UMR_MD1, U-1261, Plateforme protéomique, Aix-Marseille Université, Faculté de Pharmacie, 27 Boulevard Jean Moulin, 13005 Marseille, France. [4]These authors contributed equally: Muriel Masi, Julia Vergalli, Jean-Marie Pagès. ✉email: jean-marie.pages@univ-amu.fr

In recent years, the clinical use of β-lactams is turning ineffective due to the spread of multidrug-resistant isolates of members of the so-called ESKAPE group that comprises *Enterococcus*, *Staphylococcus*, *Klebsiella*, *Acinetobacter*, *Pseudomonas*, and *Enterobacter* species[1,2]. Although this can be mainly due to the overexpression of chromosomic and/or plasmidic (acquired) β-lactamases, a number of studies also report multidrug-resistant isolates of *Enterobacteriaceae* that exhibit a reduced membrane permeability due to the loss or functional mutations of porins[3–27]. Alone or in combination with other mechanisms, porin defects contribute to noticeable levels of resistance[28–30]. *Klebsiella aerogenes* (formerly *Enterobacter aerogenes*), *Enterobacter cloacae* and *K. pneumoniae* produce two main classical porins—Omp35 and Omp36, OmpE35 and OmpE36, OmpK35, and OmpK36, respectively—which are orthologues of OmpF and OmpC of *Escherichia coli*[31]. β-lactams, which are generally small and hydrophilic molecules, use porin channels as a preferred route for translocating across the outer membrane into the periplasm, where they interact with target penicillin-binding proteins (PBPs). Thus, it is essential to understand the properties of porin channels in order to elucidate drug resistance associated to porin loss/modifications.

Although the separation and the identification of porins is often not straightforward, the loss or alterations of OmpF orthologues in clinical isolates of *Enterobacteriaceae* seems to be more often reported than that of OmpC orthologues, suggesting that OmpF-type channels are the more relevant ones[32]. The less permeable feature of OmpC first suggested a narrower pore size[33,34]. However, the comparison of OmpC and OmpF did not show a notable difference in the pore size of the channels but rather pointed to enhanced negatively charged residues in the pore interior of OmpC compared to OmpF[34,35]. This was in line with the previous observation that OmpC is more cation-selective than OmpF[36] and could also account for the low permeability of OmpC for anionic β-lactams[37–39]. Recent results obtained either by replacing ten titratable residues that differ between OmpC and OmpF or by changing environmental parameters such as the ionic strength led to a substantial increase of OmpC permeability in intact cells, thus confirming the importance of the charge distribution at the pore lining as the most critical parameter that physiologically distinguishes OmpC from OmpF[40].

The structures of OmpF- and OmpC-type porins of *K. aerogenes*, *E. cloacae* and *K. pneumoniae* have been solved by X-ray crystallography[41]. Despite structural similarities, the calculated electric fields show substantial differences: OmpC orthologues possess a smaller pore radius, lower conductance, higher cation-selectivity, and lower intensity of the transversal electric field compared to OmpF orthologues. Predicted permeabilities and liposome swelling rates showed that the major limitations for solutes diffusion through enterobacterial porins are determined by their atomic charges (net charge and dipole) and their size (minimal projection area)[41].

Here, the use of real-time susceptibility assays showed that among cephalosporins, ceftazidime (CAZ), which exhibits a net negative charge[42], efficiently killed cells expressing OmpF orthologues; while others such as cefepime (FEP) or cefotaxime (CTX), killed cells expressing OmpF or OmpC-type porins indifferently. This indicated that CAZ prefers entering the cells through OmpF-type porins, as further confirmed by quantifying the drugs accumulated inside bacteria by using LC–MS/MS. These data were complemented with experiments on isolated systems. We adopted an original fluorescent artificial receptor-based membrane assay (FARMA)[43,44] to monitor in real-time antibiotic transport across *E. coli* OmpF or OmpC reconstituted in proteoliposomes. We also reconstituted individual porins into planar lipid bilayers and measured the reversal potential caused by the difference between the

electrophoretic mobility of the antibiotics and their counterion across the channel. Both systems showed one-two order of magnitude faster translocation, suggesting that lipopolysaccharide (LPS) is a substantial barrier to antibiotic permeation in intact bacteria.

Overall, our data can explain the decreased susceptibility to some but not all cephalosporins observed in clinical strains of enterobacteria, which often only express OmpC-type porins (reviewed in[28]). More importantly, one could predict that the extensive use of CAZ in combination with avibactam to fight against carbapenem-resistant strains could favor the emergence of antibiotic resistance phenotypes due to the loss of OmpF-type (CAZ-permeable) porins.

## Results

**β-lactam permeation through OmpF and OmpC-type porins.** Changes in porin expression have a noteworthy impact on the resistance of clinical strains of *Enterobacteriaceae* to β-lactams. OmpF and OmpC orthologues from *E. cloacae*, *K. aerogenes* and *K. pneumoniae* were expressed in a porin less derivative of *E. coli* W3110 (W3110ΔFC) from a pBAD vector under the control of an arabinose-inducible promoter (Supplementary Fig. 1). Permeation of seven β-lactam antibiotics, including two carbapenems (ertapenem, ETP, and meropenem, MEM), three cephalosporins (ceftazidime, CAZ; cefepime, FEP; and cefotaxime, CTX) and two penicillins (piperacillin, PIP and ticarcillin, TIC), was first approached by monitoring bacterial metabolic activity in the presence of resazurin[45]. In the absence of antibiotics, growing bacterial cells reduce resazurin into resorufin, which emits fluorescence at 590 nm (Supplementary Fig. 2a, b). The addition of twice the MIC concentration rapidly inhibits resazurin reduction in the strains expressing porin orthologues but not the empty plasmid (Supplementary Fig. 2c, d). Starting with an inoculum of ~$2 \times 10^6$ cells/well, the optimal time point for fluorescence readings to quantify the metabolic shutdown was around 200 min. When plotted as % of metabolic inhibition, this cancels out target potencies of the drugs and allows comparison of relative antibiotic permeation through individual porins. The data show that OmpF orthologues provide similar susceptibility to all tested β-lactams (~60% metabolic inhibition) (Fig. 1a), while OmpC orthologues have a reduced susceptibility for some of them (CAZ, 6%; TIC, 33% compared to ~60% metabolic inhibition for FEP) (Fig. 1b). Because active efflux is another variable in determining the antibacterial activity of drugs including that of β-lactams[46], the assay was also performed in the presence of 10 μM CCCP, which is reported to collapse the proton motive force required for efflux. Inactivation of efflux has no effect on the metabolism of the cells (Supplementary Fig. 3). Therefore, we concluded that the observed phenotypes are solely due to the expression of OmpF or OmpC orthologues. *E. coli* OmpF and OmpC channels can be used as model porins. When transformed with a plasmid that constitutively expresses an AmpC β-lactamase, the metabolic activity of both W3110ΔompC (OmpF+) and W3110ΔompF (OmpC+) was not affected by the addition of CAZ, as the antibiotic is efficiently degraded in the bacterial periplasm. The antibacterial activity of CAZ was restored in the presence of tazobactam and clavulanic acid, which inhibit AmpC, in W3110ΔompC but not in W3110ΔompF (Fig. 1c). Taken together, these results show that OmpC channels are permeable to some but not to all β-lactams.

**Intracellular accumulation and killing rate of CTX, CAZ, and FEP in *E. coli* expressing OmpF or OmpC.** CTX, CAZ (third generation) and FEP (fourth generation cephalosporin) are chemically very close (Supplementary Fig. 4)[42]. Their core structure is modified at $R_1$ and $R_2$ positions, with the latter containing an

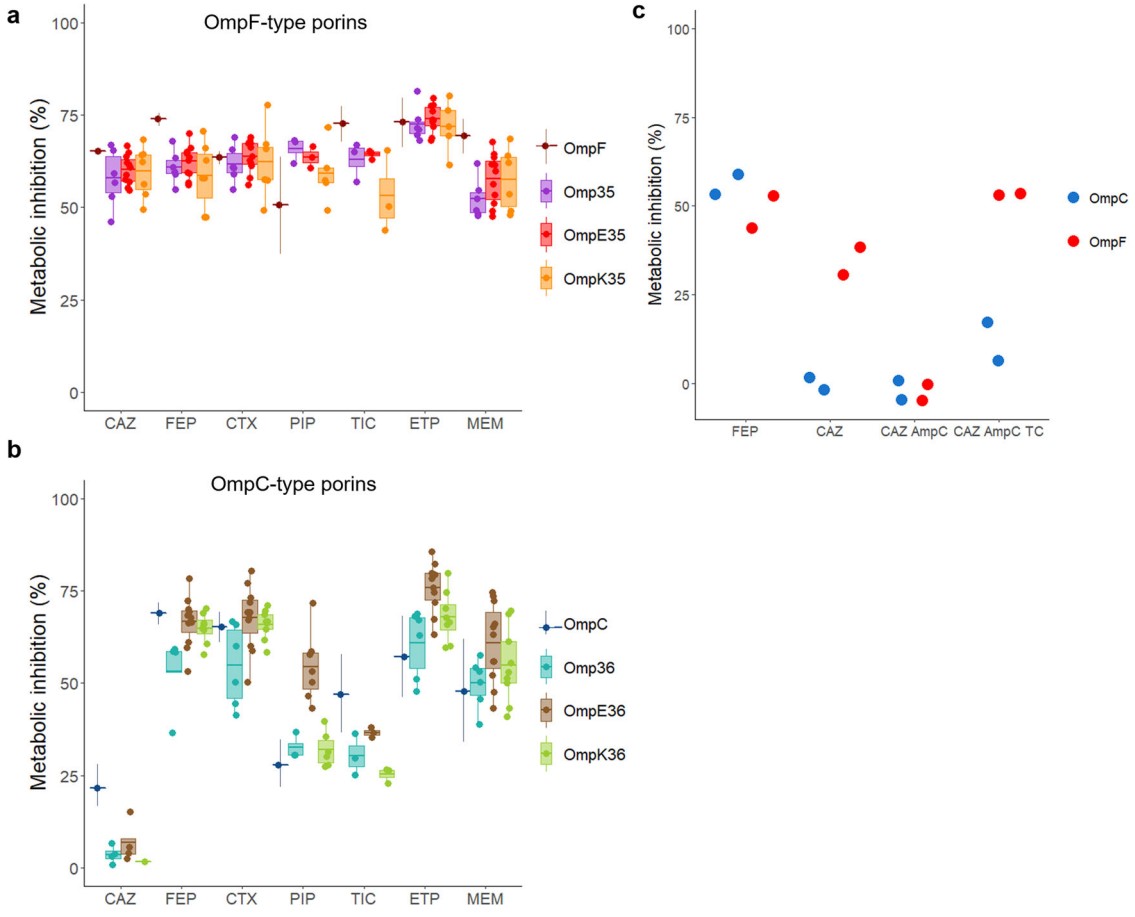

**Fig. 1 Comparison of the metabolic inhibition of *E. coli* cells expressing individual OmpF or OmpC orthologues in the presence of β-lactam antibiotics by using a resazurin-reduction-based assay.** Actively metabolizing bacterial cells are able to reduce blue resazurin into red resofurin, which emits fluorescence at 590 nm. The experiment was performed on a microtiter plate and fluorescence was measured every 10 min with $\lambda_{ex} = 530$ nm and $\lambda_{em} = 590$ nm. Inhibition of resazurin reduction in the presence of each antibiotic was translated into % metabolic inhibition. **a** W3110Δ*FC* expressing OmpF-type porins; **b** W3110Δ*FC* expressing OmpC-type porins. **c** W3110 expressing OmpF or OmpC and derivatives expressing constitutive AmpC β-lactamase from a plasmid were exposed to FEP or CAZ in the absence or in the presence of tazobactam and clavulanic acid (TC, 4 µg/ml each). Data plotted in Fig. 1a, b with strains expressing OmpF and OmpC are the mean (±SD) of three independent experiments. Results shown with boxplot in Fig. 1a, b were obtained from $n = 6, 10, 7, 6, 5, 4$ (CAZ); 6, 11, 7, 4, 11, 8 (FEP); 6, 11, 8, 6, 11, 9 (CTX); 3, 3, 5, 3, 7, 6 (PIP); 3, 3, 3, 3, 3, 3 (TIC); 6, 10, 5, 6, 10, 8 (ETP); 6, 11, 6, 6, 11, 9 (MEM) independent assays with strains expressing Omp35, OmpE35, OmpK35, Omp36, OmpE36, and OmpK36, respectively. Boxplots shown in Fig. 1a, b indicate the range from the first quartile to the third quartile of the distribution with boxes, the medians are indicated by a line across the boxes, and the whiskers extend to the most extreme data points. Results plotted in Fig. 1c were obtained from $n = 2$ independent experiments.

aminothiazole moiety. Therefore, we next performed accumulation of CTX, CAZ, and FEP in both W3110Δ*ompF* and W3110Δ*ompC* by using a previously validated protocol using LC–MS/MS to quantify the intracellular concentration of each compound[47,48]. The accumulation of all three antibiotics increased over the time to reach a plateau after 30 min of incubation (Fig. 2 and Supplementary Fig. 5), which reflects facilitated diffusion through the porin channels. However, the permeation rate of CAZ is more effective in the strain expressing OmpF than that expressing OmpC in the first 5 to 15 min of drug exposure (Fig. 2a). In contrast, no significant variation in the permeation rates of FEP and CTX was observed between the two strains (Fig. 2b and Supplementary Fig. 5a). These results demonstrate that CAZ has a clear preference for permeating through OmpF over OmpC, while FEP and CTX have no preference.

The same samples were used for CFU monitoring in order to associate intracellular drug concentrations to drug killing rates. Figure 3 shows the rate of killing curves of W3110Δ*ompF* and W3110Δ*ompC* in the presence of CAZ or FEP. In line with the accumulation data, CAZ shows a difference in antibacterial activity depending on whether the cells express OmpF or OmpC in the early

times of incubation (i.e., after 15 min of drug exposure, dead bacteria account for 60% of OmpF+ vs. 20% of OmpC+). This difference decreases over the time, reflecting the steady state of internal concentration (Fig. 3a). In comparison, FEP and CTX kill bacteria regardless of the type of porin expressed (i.e., after 15 min of drug exposure, dead bacteria account for 40% of OmpF and 40% for OmpC producer) (Fig. 3b and Supplementary Fig. 5b). When plotted versus intracellular drug concentrations, drug killing rates showed a very good correlation (Fig. 4).

Polymyxin B nonapeptide, which disrupts the outer membrane integrity, allows fast and non-selective uptake of cephalosporins in porin-less *E. coli* strains[49]. Here, Polymyxin B nonapeptide efficiently restored the accumulation rate of CAZ in W3110Δ*FC* to the level observed in the strain expressing OmpF (Supplementary Fig. 6). This confirms that the channel properties (steric, charge, flexibility, and volume) of OmpF but not of OmpC-type porins confer optimal permeation rates for CAZ.

**Permeation of cephalosporin through isolated porins OmpF and OmpC in LUVs.** We adopted the recently developed FARMA (fluorescent artificial receptor-based membrane assay)

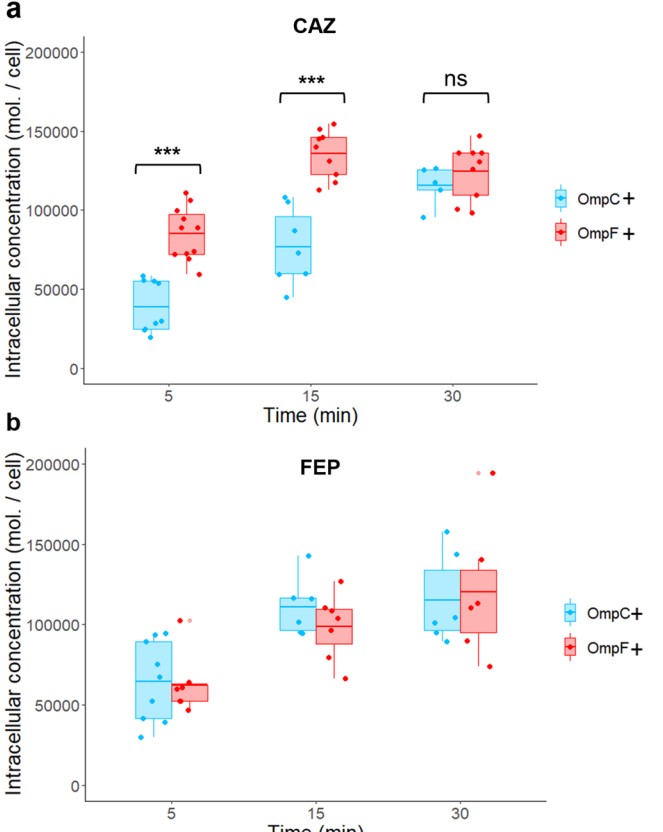

**Fig. 2 Intracellular accumulation of CAZ and FEP in *E. coli* W3110Δ*ompC* and W3110Δ*ompF*.** Intracellular concentrations were obtained from incubation of bacteria with CAZ (**a**) or FEP (**b**) at 16 μg/ml. Bacterial suspensions were sampled at 5, 15, and 30 min for CFU determination and LC–MS/MS analysis of the intracellular concentrations. Accumulation is reported in a number of antibiotic molecules per CFU. Results were obtained from $n = 3$ independent experiments performed in triplicate. Results are shown with boxplots where the boxes range from the first quartile to the third quartile of the distribution, the median is indicated by a line across the box, and the whiskers extend to the most extreme data points. ANOVAs with Tukey's post hoc tests were used to determine differences between the two strains (***$P < 0.001$; **$P < 0.01$; *$P < 0.05$). Concentrations of CAZ in W3110Δ*ompC* and W3110Δ*ompF* are significantly different at 5 and 15 min ($F(1,18) = 38.35$, $P = 7.4 \times 10^{-6}$ and $F(1,14) = 35.34$, $P = 3.6 \times 10^{-5}$). Concentrations of FEP accumulated in W3110Δ*ompC* and W3110Δ*ompF* are not significantly different.

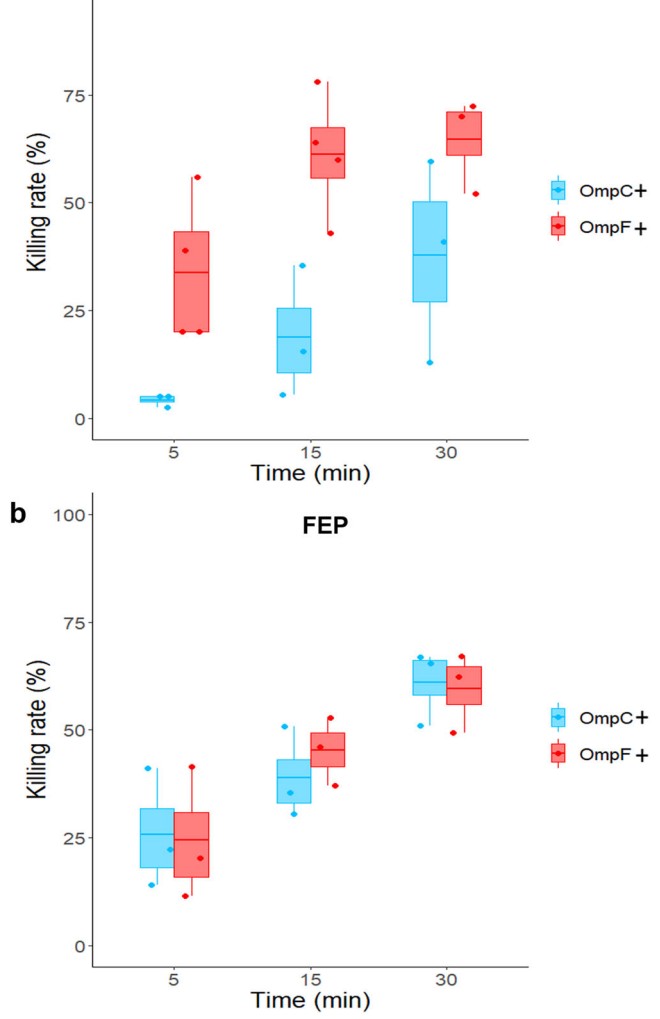

**Fig. 3 Killing rates of CAZ and FEP during accumulation in *E. coli* W3110Δ*ompC* and W3110Δ*ompF*.** Bacteria were incubated with 16 μg/ml of CAZ (**a**) or FEP (**b**), and bacterial suspensions were sampled at 5, 15, and 30 min for CFU determination. The data were obtained from $n = 3$ independent experiments performed in duplicate or triplicate. Results are shown with boxplots where the boxes range from the first quartile to the third quartile of the distribution, the median is indicated by a line across the box, and the whiskers extend to the most extreme data points. Kruskal–Wallis tests were used to determine differences in killing rates between the two strains. Killing rates of CAZ are significantly different between W3110Δ*ompC* and W3110Δ*ompF* ($P = 8.4 \times 10^{-7}$), while no significant difference was observed for the FEP killing rates.

method, that can be used for direct time-resolved monitoring of direct as well as porin-assisted permeation of antibiotics across isolated porins (Supplementary Fig. 7a)[43,44,50]. Since the method can be used for both low as well as high-molecular-weight analytes, it should be equally applicable for fluorescence-based monitoring of antibiotic uptake.

Briefly, the assay is based on the encapsulation of a fluorescent artificial receptor (FAR) consisting of a macrocyclic receptor and a fluorescent dye into large unilamellar vesicles (LUVs). For monitoring the uptake of the antibiotics CAZ and FEP, we identified the strongly fluorescent complex between cucurbit[8] uril and *N,N*-dimethyl-2,7-diazapyrenium, CB8/MDAP, as the best choice, which corresponds to FAR1 from the original study[43]. The CB8/MDAP receptor forms a non-fluorescent ternary complex with antibiotics, specifically CAZ, FEP and CTX, and sizable millimolar binding constants were determined by fluorescence titrations (Supplementary Fig. 7b, c). When the fluorescent receptor is encapsulated in LUVs, the porin-assisted

uptake of antibiotics can be monitored in real-time through switch-off fluorescence response[51] (Supplementary Fig. 7a).

We used LUVs with an average diameter of 130 nm and with a homogeneous porin/vesicle distribution of about one to ten trimeric porins. Control experiments with porin-containing LUVs without antibiotics confirmed that the CB8/MDAP reporter pair did not escape through the reconstituted porin channels (Supplementary Fig. 7d). Vice versa, controls with porin-free LUVs demonstrate that the addition of antibiotics did not show a notable difference in MDAP emission fluorescence, while that of a fast-permeating analyte such as tryptophane amide resulted in rapid fluorescence quenching (Supplementary Fig. 7e). Upon reconstitution of either OmpF or OmpC, CAZ, FEP, and CTX permeated through the porins into the LUV lumen, as

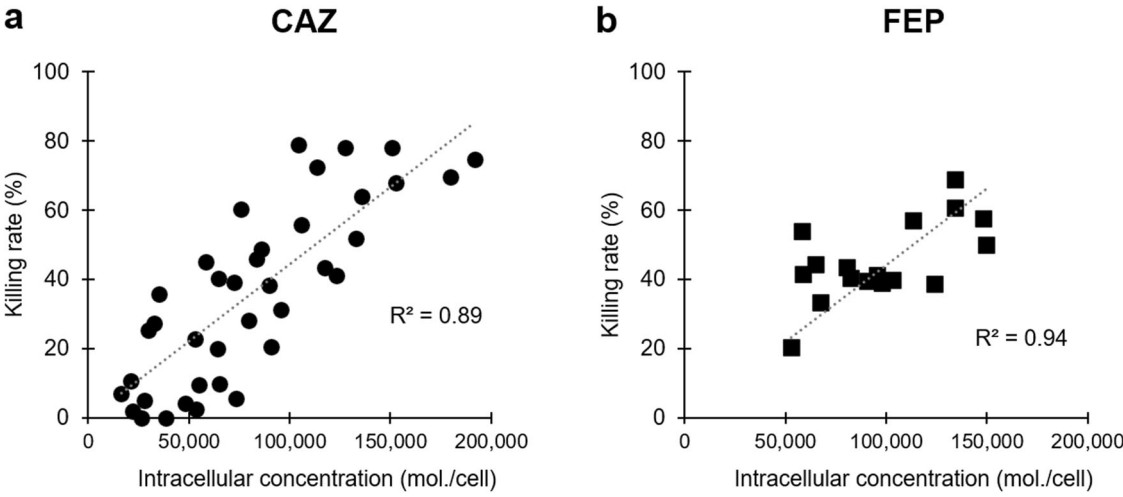

**Fig. 4 Correlation between killing rate and intracellular accumulation of CAZ and FEP.** Intracellular accumulation of CAZ (**a**) and FEP (**b**) were measured at different time points during incubation with W3110Δ*ompC*, W3110Δ*ompF*, and W3110Δ*CF*. The corresponding coefficients of determination ($R^2$) were 0.89 ($n = 38$) and 0.94 ($n = 17$), respectively.

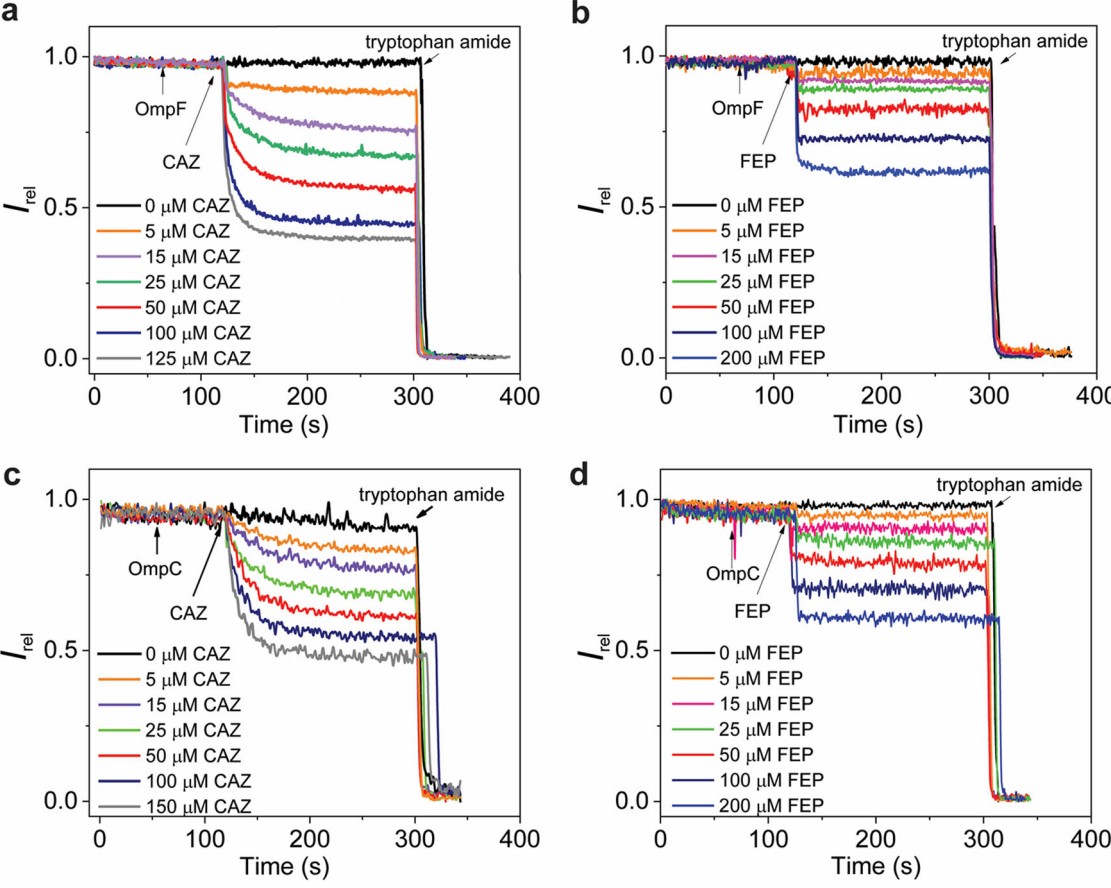

**Fig. 5 Real-time permeation kinetics of CAZ and FEP through OmpF or OmpC porin channels as monitored by the FARMA method.** Shown is relative fluorescence intensity ($\lambda_{ex} = 339$ nm; $\lambda_{ex} = 422$ nm) of CB8/MDAP-loaded LUVs (15 μM phospholipids in 10 mM Hepes, pH 7.0) upon addition of 45 nM OmpF (**a**, **b**) or OmpC (**c**, **d**) at $t = 60$ s and, after reconstitution, of a given concentration of CAZ (**a**, **c**; 0–150 μM) or FEP (**b**, **d**; 0–200 μM) at $t = 120$ s. 25 μM of tryptophan amine was added for calibration at $t = 600$ s.

indicated by the time-resolved decrease of fluorescence (Fig. 5 and Supplementary Fig. 8). Expectedly, increasing antibiotic concentrations resulted in both, faster kinetics and a larger fluorescence decrease at a given porin concentration. The following conclusions can be drawn: (i) Permeation of FEP through both porins occurs very fast, near the limit of time resolution, as the fluorescence plateau is reached within 10 s even at low antibiotic concentrations. (ii) The permeation of CAZ is at least one order of magnitude slower than that of FEP and comparable for both porins, with the plateau being reached within ca. 100–200 s. Whether the kinetics of CAZ differs between the two porins cannot be convincingly deduced because

the amounts of reconstituted active channels per LUV may vary to an unknown amount. To assess the flux in absolute terms, electrophysiological experiments can be employed, in which the conditions can be adjusted such that only a single porin is embedded in a lipid bilayer.

**Flux quantification of cephalosporin through single porins in planar lipid membranes.** We previously investigated the permeation of antibiotics through porin channels by analyzing the ion current fluctuations induced by the presence of small antimicrobial compounds[52]. However, these often do not cause detectable changes when passing through porins. In order to circumvent this problem, we applied another approach to characterize the transport of charged antibiotics by applying a concentration gradient and measuring the reversal potential (i.e., the electrostatic potential caused by an uneven electrophoretic diffusion of cations vs. anions across the porin channel)[53–55]. Here, multiple OmpF or OmpC channels were reconstituted into a planar lipid bilayer in symmetric buffer conditions and the ion current vs. applied voltage was measured (Supplementary Fig. 9). It is important to note that under symmetric conditions, zero applied voltage causes no ion current, while a concentration gradient of pure 50/30 mM NaCl creates a reversal potential of 13 and 16 mV for OmpF and OmpC, respectively. As expected, this difference reflects the electrostatic properties of the porin channels, with the pore interior more negative in OmpC than in OmpF. In a similar manner, we performed concentration gradients with 50/30 mM CAZ (16 mV for OmpF and 18,6 mV for OmpC) or CTX (18 mV for OmpF and 19 mV for OmpC, see Table 1). The obtained I/V curves were analyzed using Goldman–Hodgkin–Katz (GHK) ion current equation to obtain the permeability ratio between cations and anions (Table 1). Combined with single channel conductance, one can estimate the contribution of the individual ions and extrapolate it to 30 μM. We finally estimated a permeation rate for anionic CAZ of 1000 molecules/sec/monomer across OmpF vs. 500 molecules/sec/monomer across OmpC (Table 1). Slightly lower values were obtained for CTX permeating with 600 molecules per second per monomer of OmpF and 400 molecules through OmpC. Unfortunately, permeation of FEP could not be tested by this method as it carries a zero net charge, but the FARMA assays (Fig. 5) demonstrate much faster uptake kinetics for FEP than for CAZ (and CTX, see Supplementary Fig. 8). The fast kinetics for FEP points to an essentially unhindered diffusion through both porin, while the slower rate of CAZ suggests a sizable barrier, which appears to be slightly lower for OmpF than for OmpC, potentially accounting for the observed selectivity.

**Table 1 Multi-channel reversal potential experiments.**

| Substrate (Concentration gradient Bi-ionic cis-trans 80-30) | Pore | $V_{rev}$ (mV) | Permeability ratio (cation:anion) |
|---|---|---|---|
| Sodium chloride | OmpF | 13 ± 5 | 3.5:1 |
| Ceftazidime -Sodium | OmpF | 16 ± 4 | 5.2:1 |
| Cefotaxime -Sodium | OmpF | 18 ± 4 | 8:1 |
| Sodium chloride | OmpC | 16 ± 2.5 | 5.2:1 |
| Ceftazidime -Sodium | OmpC | 18.6 ± 3 | 8:1 |
| Cefotaxime -Sodium | OmpC | 17.1 ± 4.1 | 7:1 |

Bi-ionic reversal potential $V_{rev}$ (±SD, $n = 3$) is needed to obtain zero current for a concentration gradient of 80 mM/30 mM of the respective antibiotic only. Using Goldmann–Hodgkin–Katz equation gives the permeability ratio $P_{anion-}/P_{Na+}$ of sodium chloride. cefotaxime (CTX⁻) and ceftazidime (CAZ⁻) for the ion current flux through OmpF and OmpC.

## Discussion

The global spread of carbapenemases has compromised the use of carbapenems, while the combination of cephalosporins with new β-lactamase inhibitors such as avibactam represents an attractive therapeutic alternative[56]. A major challenge to the development of antimicrobials has to do with their permeation across the outer membrane of Gram-negative bacteria. Several approaches have been used to study small molecule permeability including enzymatic detection of β-lactams entering the periplasmic space[32,37,38,40,57], calculation of relative permeabilities with liposome swelling assays[38,41], determination of compound accumulation with mass spectrometry[58,59], channel permeation by electrophysiology[60–62], and molecular dynamics simulations[41,62,63]. Among these, pioneer studies from H. Nikaido and coworkers allowed the determination of influx rates and helped initiate relationship analyses between the properties of the porin channels and the molecular determinants of β-lactams, such as charge, size or hydrophobicity[32,37–40]. Accordingly, cells expressing OmpF orthologues are more susceptible to CAZ than the ones expressing OmpC orthologues in a killing assay, while no difference was observed when cells were exposed to FEP regardless of the type of porin expressed[41]. Are OmpF and orthologues specific for CAZ permeation? This question was raised by L. K. Siu back in 2001[64] after Rasheed et al. postulated that OmpK35 was specific for CAZ translocation in K. pneumoniae[65]. Since then, independent studies using susceptibility assays[66] or electrophysiology[60] have reported the CAZ preference to diffuse through OmpF-type channels. In this view, a detailed examination of OmpF versus OmpC permeability is really important to understand the drug susceptibility in Enterobacteriaceae. Herein, we provide a clear comparative analysis of CAZ and FEP translocation in intact cells and reconstituted membrane assays.

Drug accumulation in E. coli expressing individual porins during the early incubation times was quantified by LC–MS/MS. Approximately $9 \times 10^4$ molecules of CAZ were internalized after 5 min exposure time of cells expressing OmpF. In comparison, only $3 \times 10^4$ molecules of CAZ were counted in OmpC-producing cells. These amounts correspond to the translocation of 300 molecules per second per bacteria expressing OmpF versus 100 molecules per second per bacteria expressing OmpC. A steady-state accumulation level of CAZ is observed after 15 min of drug exposure, conjointly to the plateau of killing rate observed with cells expressing OmpF, while twice as much time was required for cells producing OmpC. In contrast to CAZ, FEP accumulates similarly in OmpF and OmpC expressing cells at a rate of ~425 molecules/sec/bacterial cell. For CTX, the accumulation profile is quite similar to that of FEP. From these results, it is likely that charges play a key role in drug uptake. Interestingly, the $IC_{50}$ of CAZ and FEP are similar for PBP1a and 1b, while CAZ does not bind as well as FEP to PBPs 2–4[67,68]. With a total number of PBP2 and PBP4 estimated at about 370 molecules/cell[69], it is clear that the periplasmic concentration of CAZ must reach higher values than FEP to exert an effective antibiotic activity.

Quantification of the antibiotic permeation rates through reconstituted porin channels supports a different uptake of CAZ with respect to E. coli OmpF and OmpC. Overall, the one-two order of magnitude slower translocation rates observed in intact bacteria compared to artificial lipid systems suggest that LPS could be a substantial barrier to antibiotic permeation. This could be tested with outer membrane vesicles containing LPS[70]. In addition, the absence of antibiotics targets (i.e., PBPs) might reduce further the uptake kinetics.

Importantly, with the emergence of carbapenem-resistant strains and the recent use of the CAZ-avibactam combination, it is necessary to relate these data to clinical aspects. Several studies reported that during the clinical use of cephalosporins to

treat infections caused by *K. pneumoniae, K aerogenes*, a step-by-step process occurs altering the porin expression pattern. In these isolates, a change in the porin balance (expression of OmpK35 is stopped while the OmpK36 level is held) is observed at the beginning of antibiotic therapy, which is rapidly moving towards a general porin deficiency (loss of both OmpK35 and OmpK36) often associated with efflux pump overexpression (reviewed in ref. [28]). Such changes contribute to the enterobacterial adaptive response to antibiotic therapy and also may explain the hetero-resistance observed in bacterial strains isolated in different infectious sites[71]. This has also been observed when *E. coli* cells are exposed to various antibiotics[72] and indicates that sophisticated adaptive response includes an efficient interplay of sensors that rapidly detect the β-lactam concentration in the periplasm and genetic regulators that down-regulate the porin expression[28,73].

## Methods

**Bacterial strains and reagents.** All strains used in this study are derived from *E. coli* K12 W3110. Individual outer membrane porin deleted (W3110ΔompF and W3110ΔompC) and porinless (W3110 ΔompF ΔompC) strains were gift from M. G. P. Page. W3110ΔFC was transformed with the empty pBAD24 vector, or recombinant derivatives containing *K. aerogenes omp35* or *omp36*, *E. cloacae ompE35* or *ompE36*, or *K. pneumoniae ompK35* or *ompK36*. All strains were cultured in Mueller Hinton II Broth or Luria Broth supplemented with ampicillin (100 μg/ml), and protein expression was induced with 0.02% L-arabinose (Sigma) for 2 h at 37 °C. Carbapenems including ertapenem (ETP) and meropenem (MEM), were purchased from Sequoia Research Products Ltd. (United Kingdom); cephalosporins including ceftazidime (CAZ), cefotaxime (CTX), and cefepime (FEP); and penicillins including piperacillin (PIP) and ticarcillin (TIC) were from Sigma. Anhydrous pentane, hexadecane and hexane was purchased from Carl Roth GmbH, Co. 1,2-diphytanoyl-*sn*-glycero-3-phosphocholine (DPhPC), 1-palmitoyl-2-oleoyl-*sn*-glycerol-3-phosphocholine (POPC), and 1-palmitoyl-2-oleoyl-*sn*-glycerol-3-phospho-L-serine (POPS) were purchased from Avanti Polar Lipids (Alabama, USA). All solutions were prepared with 18.2 MΩ cm Millipore-grade water unless otherwise noted.

**Preparation of crude outer membrane fractions and outer membrane protein analysis.** The crude envelope fractions were prepared with a cell disruptor (Constant Systems) followed by two steps of centrifugation (8000 × *g*, 20 min, then 100,000 × *g*, 60 min). These were solubilized in 0.3% Sarkosyl to remove inner membrane proteins. The Sarkosyl-insoluble fraction containing the outer membrane fragments was collected by ultracentrifugation and resuspended in 20 mM HEPES-NaOH buffer (pH 7.2). All samples were diluted in Laemmli buffer and heated for 5 min at 100 °C before loading. Samples corresponding to 0.2 OD units were separated on SDS-PAGE. Proteins were either visualized after straining with Coomassie Brilliant Blue R250 or transferred onto PVDF Blotting membranes (Bio-Rad). Immunodetection of OmpF and OmpC orthologues was performed on separated membranes with polyclonal rabbit antibodies directed against the denatured monomers of *E. coli* OmpF (α-OmpFd, 1:5000) or OmpC (α-OmpCd, 1:10,000). Goat anti-rabbit HRP-conjugated secondary antibodies and Clarity Max ™ ECL Western Blotting Substrates (Bio-Rad) were used for detection. Protein bands were visualized using a molecular imager ChemiDoc-XLS (Bio-Rad).

**Determination of minimum inhibitory concentrations.** Minimum inhibitory concentrations (MICs) were determined by using 96-well microtiter plates and a standard twofold microdilution method in Mueller Hinton II Broth. Cultures were grown, and ~2 × 10^5 cells were inoculated to each well. Results were read after incubation at 37 °C for 18 h. MICs were obtained three times during three biologically independent assays. Results are shown in Supplementary Table 1.

**Resazurin-reduction-based antibiotic uptake assays.** Cultures were exponentially grown and then diluted to 10^7 cells/ml in fresh Luria Broth supplemented with 0.02% L-arabinose, β-lactamase inhibitors tazobactam and clavulanic acid (4 μg/ml each) to inhibit the activity of the plasmidic AmpC and 10% (v/v) of Cell-Titer Blue® Cell-Titer Viability Reagent (Promega). About 190 μl of these mixtures were added to separated wells of 96-well microplate with black sides and clear bottom (Costar catalog no. 3094) containing 10 μl of 20× antibiotic solutions. For each antibiotic, the final concentration in the wells was defined as to the maximal concentration that yielded negligible metabolic inhibition for W3110ΔFC transformed with the empty vector, (i.e., ETP, 0.125 μg/ml; MEM, 0.125 μg/ml; CAZ, 0.25 μg/ml; FEP, 0.125 μg/ml; CTX, 0.125 μg/ml; PIP, 2 μg/ml; TIC, 8 μg/ml). Control wells also contained cells with resazurin but no antibiotic and resazurin with antibiotics without cells. Fluorescent signals of resorufin were measured with a TECAN Infinite Pro M200 spectrofluorometer (λex = 530 nm and λem = 590 nm).

Kinetic readings were taken at 37 °C every 10 min for 300 min. % of metabolic inhibition for each strain exposed to each antibiotic was calculated from the measured difference of relative fluorescence units (RFU) in the presence ($RFU_{ATB}$) as compared to in the absence ($RFU_{MAX}$) of antibiotic as illustrated in Supplementary Fig. 2. Experiments were performed 2 (Fig. 1c), 3, or 4 (Supplementary Fig. 3), and 3 to 11 (Fig. 1a, b) independent times.

**Accumulation assay, LC–MS/MS analysis, and killing rate.** An overnight culture grown in Luria broth was diluted into fresh medium (1/100; v/v) and grown at 37 °C with shaking to an optical density ($OD_{600}$) of 0.6. The bacteria were pelleted at 3000 × *g* for 15 min at room temperature and the supernatant was discarded. The cell pellets were resuspended in sodium phosphate buffer (50 mM $Na_2HPO_4$/$NaH_2PO_4$ pH 7.0) supplemented with 2 mM $MgCl_2$ to a final $OD_{600}$ of 6. In glass culture tubes, bacterial suspensions were incubated for 5, 15, or 30 min at 37 °C without and with CAZ, FEP or CTX at 16 μg/ml[47]. Polymyxin B nonapeptide was added at 102.4 μg/ml (MIC/10) during incubation of W3110ΔFC without and with CAZ. At each time point, 800 μl of suspensions were collected, diluted with 1.1 ml buffer and bacteria were immediately pelleted by centrifugation (6000 × *g* for 5 min at 4 °C). The supernatants were removed by pipetting and bacterial cells were lysed in 500 μl Glycin-HCl pH 3.0 overnight at room temperature[47]. Bacterial cell lysates were then centrifuged for 15 min at 9000 × *g* at 4 °C, the supernatants were collected, and 20 μl of supernatant was mixed with 60 μl of Glycin-HCl pH 3.0 before analysis by LC–MS/MS. Bacterial suspensions incubated without compounds were used to determine the calibration curves of CAZ, FEP, and CTX by LC–MS/MS (see below). All compounds were ≥95% pure. In parallel, suspensions were sampled at each time point of the accumulation assay for CFU determination. Suspensions were consecutively diluted in fresh medium, spread on Petri dishes and incubated overnight at 37 °C. CFUs were then used to determine intracellular concentrations as well as killing rates. Accumulation and killing assays in OmpF+ and OmpC+ strains with CAZ or FEP were carried out in triplicate during three independent assays. Accumulation and killing assays in OmpF+ and OmpC+ strains with CTX were carried out in duplicate during two independent assays. Accumulation of CAZ in the porinless strain was carried out in triplicate during one assay.

**Mass spectrometry determination.** The chromatographic separation was performed on a Phenomenex Kinetex column XB-C18 (2.6 μm 75 × 2.1 mm) protected with a cartridge pre-column $C_{18}$ 2.6 μm Phenomenex (4 × 3.0 mm) at 35 °C with the flow rate of mobile phase at 0.55 ml/min. The mobile phase consisted of (A) 0.1% formic acid and ammonium acetate 5 mM in water and (B) 0.1% formic acid in methanol. The gradient elution program was applied as follows: at 0.50 min, phase (B) increased to 30% in 2.0 min, then increased to 98% in 3.50 min, and was held for 0.50 min before returning to initial conditions in 0.80 min. The column was re-equilibrated 1.20 min before a new analysis. The sample injection volume was 5 μL. The Nexera-UPLC system was coupled to an 8040 Triple Quadrupole Mass Spectrometer (Shimadzu, France) equipped with an electrospray ionization (ESI) source. The MS/MS detection was performed in mode ESI+ under the following conditions: dry gas (nitrogen) temperature, 160 °C; dry gas flow 16 L min^−1; sheath gas (nitrogen) temperature, 350 °C; sheath gas flow, 12 L min^−1; nebulizer pressure, 35 psi; nozzle voltage, 1500 V; capillary voltage, 4000 V; fragmentation voltage, 380 V. The values of collision energy, transitions, and data acquisition time segments for the multiple-reaction monitoring (MRM) mode are: CAZ precursor ion 547.1 and products ions 468.0; 396.1 with collision energy 12ev/18ev, retention time (RT):2.038 min; FEP precursor ion 481.2 and products ions 86.2; 396.0; 125.1, and 86.20 with collision energy 22ev/12ev/53ev, RT: 1.36 min; CTX precursor ion 456.15 and products ions 125.1; 167.20 and 396.20 with collision energy 45ev/21ev/10ev; RT: 2.72 min.

**Standard curves.** Solutions of CTX, CAZ, and FEP were prepared at 10 mg/ml in Glycin-HCl pH 3 buffer. A mixed solution was performed at 1 mg /ml. Serial dilutions were done to final concentrations of 5, 10, 20, 40, 60, 80, and 160 ng/ml. The different standard curves were performed with 20 μl of bacterial lysate incubated without compound and 60 μl standard solutions (Supplementary Fig. 10).

**Antibiotic permeation using porin-containing LUVs and FARMA.** For LUV preparation, a thin lipid film was prepared by evaporating the lipid solution (100 μl 25 mg/ml POPC and 33 μl 10 mg/ml POPS in chloroform) with a stream of nitrogen and dried under vacuum overnight. The lipid film was rehydrated (stirring 30 min at room temperature) with 1 ml buffer (550 μM MDAP, 500 μM CB8, 10 mM Hepes, pH 7.0) and subjected to 20 freeze/thaw cycles. Extravesicular components were removed by size exclusion chromatography (NAP-25 Column) with 10 mM Hepes, pH 7.0, to give CB8-MDAP LUVs. The vesicle size was confirmed by DLS (on a Malvern Instruments DTS nano 2000 Zeta-Sizer), and the phospholipid concentration was calculated by the Stewart assay[74]. For transport experiments, CB8-MDAP LUVs stock solutions were diluted with buffer (10 mM Hepes, pH 7.0) in a disposable plastic cuvette and gently stirred (total volume 2000 μL, final lipid concentration 15 μM). MDAP fluorescence was monitored at $\lambda_{em}$ = 422 nm ($\lambda_{ex}$ = 339 nm) as function of time. The absence of reporter pair in the extravesicular phase was verified by adding tryptophan (which is a non-

permeable CB8 analyte). Cephalosporin transport was evaluated by monitoring MDAP emission after the addition of porin (OmpF or OmpC, at $t = 60$ s), antibiotic (CAZ or FEP at $t = 120$ s), at the end of the experiment ($t = 600$ s) tryptophan amide (25 μM, as a membrane-permeable analyte) was added for calibration. Fluorescence intensities were normalized to factional emission as

$$I_f = \frac{I_t - I}{I_0 - I} \tag{1}$$

where $I_0 = I_t$ at porin addition, and $I_\infty = I_t$ at saturation after tryptophan amide addition. Fluorescence was measured in a Varian Cary Eclipse spectrofluorometer equipped with a temperature controller (25 °C) in 3.5 ml polymethylmethacrylate cuvettes from Sigma-Aldrich.

**Single channel conductance in the presence of antibiotics.** Small amounts (<1 μl) from a diluted stock solution ($10^{-2}$–$10^{-5}$ mg/ml) of OmpF or OmpC were added to the bilayer chamber containing the selected buffer. Typically, within a few minutes, the conductance increases stepwise, each step is interpreted as channel insertion. From the distribution of the conductance steps, we conclude on the single channel trimer conductance[75,76]. In agreement with the literature, $G_{trimer} = 272 \pm 35$ pS for OmpF and $G_{trimer} = 140 \pm 30$ pS for OmpC in 30 mM NaCl (pH 7.0) (Supplementary Table 2)[75,76].

**Multi-channel reversal potential measurements.** OmpF and OmpC were first reconstituted in a 30 mM CAZ-Na or CTX-Na buffer, compound concentration was then raised on one side (the *cis* side corresponding to the electrical ground side) to 80 mM. $I/V$ curves were recorded for all tested conditions (Supplementary Fig. 9). As cations and anions have different permeability, a shift of the zero current potential is observed called reversal potential[53]. Using the GHK equation allows to calculate the permeability ratio $P_{cation}/P_{anion}$:

$$\Delta V = \frac{RT}{F} \times \ln\left(\frac{P_{cation} \times [cation]^{cis} + P_{anion} \times [anion]^{cis}}{P_{cation} \times [cation]^{trans} + P_{anion} \times [anion]^{trans}}\right) \tag{2}$$

where $\Delta V$ is the measured reversal potential, $P_{Na}$ the permeability for $Na^+$ and $P$ the permeability for $Cl^-$, $CAZ^-$, or $CTX^-$, the universal gas constant $R = 8.3$ J/mol$^{-1}$ K$^{-1}$, and the Faraday constant $F = 9.6 \times 10^4$ C/mol$^{-1}$.

The permeability ratio for Na/ceftazidime in OmpF under bionic conditions (Table 1) was $P_{CAZ^-}/P_{Na^+} = 0.20$, whereas for/OmpC was $P_{CAZ^-}/P_{Na^+} = 0.12$. In case of the Na/cefotaxime in OmpF was $P_{CTX^-}/P_{Na^+} = 0.12$ compared to OmpC $P_{CTX^-}/P_{Na^+} = 0.1$.

**Estimation of the single channel permeability.** The conductance of a bi-ionic antibiotic solution gives the total flux of ions under a given external voltage and the permeability ratio allows to distribute the ion flux on the respective cations and anions. However, the concentration-driven flux is less obvious as both charges move in the same direction, and only the difference gives rise to an electrical signal. The reversal potential balances both fluxes and can be used to estimate the strength. Extrapolation from 30 mM to 30 μM results in the molecular flux of molecules (molecules/s). For example, to obtain an estimate for the flux of CAZ across a single OmpF monomer (molecule.s$^{-1}$), we first calculate the ion current at the reversal potential $I_{total}$ (30 mM) = $G_{single\ monomer} \times V_{rev} = (182\ pS/3) \times 16$ mV = $10^{-12}$ A or divided by a single charge const gives the total flux of ions $\Phi = 10^{-12}$ A/$1.6 \times 10^{-19}$ A s = $6 \times 10^6$ ion/s. The calculated permeability ratio gives the fraction of CAZ molecules and we further extrapolate the flux from 30 mM to 30 μM gives $\Phi = 6 \times 10^6 \times 10^{-3}/6.2 = 1000$ CAZ/s. Note that this is a very rough estimation and a more accurate estimation can be obtained using GHK for the ion current under a concentration coefficient[53].

**Statistics and reproducibility.** Statistical analyses were performed using the computing environment R (R Development Core Team, 2020). ANOVAs with Tukey's post hoc tests were used to determine differences in accumulation between OmpF and OmpC producers. $P$ values ≥0.05: not significant (ns), $P$ values between 0.01 to 0.05: significant (*), $P$ values between 0.001 to 0.01: very significant (**), $P$ values <0.001: extremely significant (***). The data normality and homoscedasticity were checked by the respective Shapiro–Wilk and Fligner–Killeen tests. The non-parametric Kruskal–Wallis test was used to determine differences in killing between the studied strains.

Metabolic inhibition assays were performed at two (Fig. 1c), three to eleven (Fig. 1a, b), three or four independent times (Supplementary Fig. 3). Accumulation and killing assays with CAZ and FEP were carried out in triplicate or duplicate (killing) during three independent assays (Figs. 2 and 3) or in triplicate during one assay for accumulation in the porinless strain (Supplementary Fig. 6). Accumulation and killing assays with CTX were carried out in duplicate during two independent assays (Supplementary Fig. 5). Pearson's Coefficients of determination were calculated with $n = 38$ (CAZ) and $n = 17$ (FEP) independent samples (Fig. 4).

**Reporting summary.** Further information on research design is available in the Nature Research Reporting Summary linked to this article.

## Data availability
All data supporting this study are available within the article and its Supplementary Information. Source data are available within the Supplementary Data 1 file. All other data are available from the corresponding author on reasonable request.

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

## Acknowledgements

The research leading to these results was partly supported by the TRANSLOCATION consortium, and it has received support from the Innovative Medicines Initiatives Joint Undertaking under Grant Agreement n°115525, resources which are composed of financial contribution from the European Union's seventh framework program (FP7/2007-2013) and EFPIA companies in kind contribution. This work was also supported by Inserm, Aix-Marseille Univ. and Service de Santé des Armées. We would like to thank J.M. Brunel, J.M. Bolla, and A. Davin-Régli for their discussions and intellectual contribution. We also thank J.A. Bafna in the earlier stage of this study for his valuable contribution to the biophysical characterization of OmpF and OmpC channels.

## Author contributions

M.M., T.S., I.G., and J.V. designed experiments, generated, and analyzed biological data; T.S. and D.L. performed mass spectrometry and data analyses; I.G. and M.W. performed electrophysiological measurements; A.B.-B. and W.M.N. designed and performed the vesicle experiments; M.W. and J.-M.P. contributed to experimental design and data interpretation; M.M., J.V., M.W., and J.-M.P. co-wrote the manuscript, with input from all the authors. The authors submitted this study in memoriam to Prof. Isabelle Artaud, deceased in March 2021, who contributed to numerous discussions regarding antibiotic transport across biological membranes.

## Competing interests

The authors declare no competing interests.
