## [Peer Review File · Communications Biology]

Reviewers' comments:

Reviewer #1 (Remarks to the Author):

In this work "Cephalosporin translocation across enterobacterial OmpF and OmpC channels, a filter across the outer membrane"), Masi, et al. investigate the effect of porins on the entry of three cephalosporins into *E. coli*. Porins are the major entry into gram-negative cells for smaller antibiotics and changes in porin expression have been repeatedly linked to antibiotic resistance. Despite this, the specific interactions of clinically relevant antibiotics (especially more recently developed antibiotics) with porins are not well understood. The authors determine that ceftazidime more active on cells producing OmpF than OmpC and show that ceftazidime can translocate OmpF faster than OmpC likely because of its negative charge. Other less negatively charged cephalosporins were equally able to kill cells in the presence of the two porins.

This paper, which is well designed and well written, should be of interest to a broad audience including those interested in clinical antibiotic resistance, gram-negative envelope biology. Clarification of several points listed below would strengthen this paper.

Comments

1. Table S1: For antibiotics affected by the ompCF deletion, the complementation of the plasmid borne copies seems to be quite weak. Please comment in the text as to why MICs for single OMP strains are similar to the vector control for many antibiotics. Also, please comment on possible mechanisms through which expression ompC would increase MIC over that of the double deletion.
2. Figure 4: The authors state that they calculated the "Pearson's correlation coefficient R^2 ". Please clarify whether the value is actually Pearson's correlation coefficient (r) or R^2 .
3. Pg 9 last sentence of 1st full paragraph: It is not clear what "IC50 of CAZ to target PBPs allow to reach higher concentration" means.
4. Second full paragraph of Pg9: Although cephalosporins are generally considered to be porin translocated, that LPS decreases the permeability of the OM to many antibiotics has been well established. This should be acknowledged in this paragraph and it should be made more clear that the open question would be whether cephalosporins are affected.

Reviewer #2 (Remarks to the Author):

The emergence of bacterial strains with antibiotic resistance is a pressing problem for society and the mechanisms of antibiotic resistance including methods to overcome this problem are topics of significant current interest. Based on cumulative literature results, the authors concluded that some antibiotic resistance may originate from the lack of outer membrane porins, which prevents certain antibiotic from reaching their intracellular target. Herein, the authors report a comparative analysis of the porin permeability of several antibiotics through the outer membrane protein F and C (OmpF versus OmpC). OmpF and OmpC were therefore expressed in *E. coli* and the cell toxicity of several antibiotics was assessed by the resazurin assay and compared to the uptake as determined by LC/MS-MS. In addition, the permeability was determined by single channel current measurements in planar lipid bilayers and a fluorescent assay in liposomes.

The authors focused on comparing the cephalosporins ceftazidime (CAZ), cefotaxime (CTX), and cefepime (FEP). The intracellular quantification by LC/MS-MS conclusively demonstrated that the intracellular concentration of CAZ is significantly lower in *E. coli* with OmpC than with OmpF. It is a reasonable conclusion that this may indeed cause an antibiotic resistance of OmpF-deficient bacteria containing only OmpC.

Ion conductance measurements with planar lipid bilayers also confirmed a lower permeation of CAZ

through OmpC than through OmpF; the permeation of CTX was also reduced through OmpC compared to OmpF, but to a lesser extent, whereas FEP permeation could not be assessed, because the molecule is uncharged. The results from the fluorescent liposome assay indicate an overall increased permeation of FEP compared to CAZ.

Overall, the results suggest that the initial hypothesis, namely that the resistance of some bacteria towards ceftazidime (CAZ) may originate from the lower permeation rates through OmpC pores, holds true.

This reviewer, however, still wonders how CTX can show a similarly strong antibiotic activity on OmpF- and OmpC-containing bacteria, when the permeation rates are also different? Another question is, why the authors did not include liposome assay results with CTX?

Reviewer #3 (Remarks to the Author):

This manuscript describes a detailed study of porin-based transport of a variety of clinical stage cephalosporin analogs into Gram-negative bacteria. Resistance to beta-lactam analogs in Enterobacteriaceae is an important health threat, especially for extended-spectrum carbapenems. This manuscript takes a variety of genetic and biophysical approaches to quantitating the dependence of several different agents on different bacterial porins. The work is carefully done and is well described. The data is presented with good reproducibility and statistical treatment. The authors determine that different beta-lactam chemotypes utilize porins differentially and suggest that potential resistance mechanism could arise from deletion or mutation of specific porins. While the work is of high quality and there is nice use of an innovative technique to directly monitor the transport of compounds across membranes, the manuscript is very descriptive in nature. There is little in the way of detailed analysis linking chemical descriptors to specific transporters, and it seems primarily that these techniques would primarily be useful in a retrospective manner. The importance of the work would be greatly enhanced if testable models could be generated and used to predict porin specificity of new agents or how to enhance porin transport by chemical modification of an existing agent. It is also not clear how important altered porin-transport is to drug resistance, this was often invoked for fluoroquinolones but clinical surveillance shows very little of this compared to gyrase mutations. Perhaps the work could be enhanced by securing a panel of cephalosporin-resistant isolated and determining if alteration to porins is seen in the wild. I think this is very nice work to study these transport phenomena but that the manuscript needs to move beyond these initial studies to merit publication in Communications Biology.

Response to reviewers' comments:

Reviewer #1 (Remarks to the Author):

In this work “Cephalosporin translocation across enterobacterial OmpF and OmpC channels, a filter across the outer membrane”), Masi, et al. investigate the effect of porins on the entry of three cephalosporins into *E. coli*. Porins are the major entry into gram-negative cells for smaller antibiotics and changes in porin expression have been repeatedly linked to antibiotic resistance. Despite this, the specific interactions of clinically relevant antibiotics (especially more recently developed antibiotics) with porins are not well understood. The authors’ determine that ceftazidime more active on cells producing OmpF than OmpC and show that ceftazidime can translocate OmpF faster than OmpC likely because of its negative charge. Other less negatively charged cephalosporins were equally able to kill cells in the presence of the two porins.

This paper, which is well designed and well written, should be of interest to a broad audience including those interested in clinical antibiotic resistance, gram-negative envelope biology. Clarification of several points listed below would strengthen this paper.

Comments

1. Table S1: For antibiotics affected by the ompCF deletion, the complementation of the plasmid borne copies seems to be quite weak. Please comment in the text as to why MICs for single OMP strains are similar to the vector control for many antibiotics. Also, please comment on possible mechanisms through which expression ompC would increase MIC over that of the double deletion.

-> We thank the referee for this comment. Table S1 shows MICs for *E. coli* W3110 deleted of ompF and ompC transformed by empty pBAD or derivative recombinant plasmid encoding ompF or ompC orthologues for several β -lactam antibiotics. A control column also shows MICs of the parental strain expressing both OmpF and OmpC transformed with the empty plasmid. This strain is highly susceptible to all tested antibiotics compared to the double mutant. However, it is worth to note that the plasmid used for expression carries AmpC as a selection marker (i) and P_{ara} promoter for protein expression (ii). (i) On one hand, a cocktail of β -lactamase inhibitors was added in MH2 broth, while ampicillin was omitted. (ii) On the other hand, we used 0.02% L-ara to induce the expression of Omp, during the 18h of incubation à 37°C. As a result, two parameters can influence the observed MIC values over time: some AmpC activity due to a loss of activity of the β -lactamase inhibitors or a loss of the plasmid (and that of porin expression) due to the absence of ampicillin and/or porin toxicity. Therefore, Table S1 is not used to compare the antibiotic susceptibility that results from the expression of an individual porin but rather as an indicator to determine the working concentrations of antibiotics used for viability testing with resazurin. Only this latter assay, carried out under the same conditions but over a much shorter time, makes it possible to compare the strains with each other as regards the permeation capacity of the porins in relation to the sensitivity to the antibiotics tested. We also noticed that Table S1 was not cited in the text: reference of Table S1 has been added in Materials and Methods, “Determination of Minimal Inhibitory Concentrations” (page 10).

2. Figure 4: The authors state that they calculated the “Pearson’s correlation coefficient R^2 ”. Please clarify whether the value is actually Pearson’s correlation coefficient (r) or R^2 .

-> We thank the referee for this comment. The coefficients shown the legend of figure 4 are coefficients of determination (R^2) but not Pearson’s correlation coefficients. We do apologize for this mistake. This has been corrected accordingly it in the manuscript (page 23).

3. Page 9 last sentence of 1st full paragraph: It is not clear what “IC50 of CAZ to target PBPs allow to reach higher concentration” means.

-> The IC50 of CAZ is higher as compared to FEP for PBP2 and 4 (CAZ-PBP2: 4-240; CAZ-PBP4: > 25 / FEP-PBP2: 0.25-0.6; FEP-PBP4: 16) [Pucci MJ, Boice-Sowek J, Kessler RE, Dougherty TJ. Comparison of cefepime, cefpirome, and cefaclidine binding affinities for penicillin-binding proteins in *Escherichia coli* K-12 and *Pseudomonas aeruginosa* SC8329. Antimicrob Agents Chemother. 1991, 35:2312-7. doi:

10.1128/AAC.35.11.2312.]. The sentence has been rephrased as to: “With a total number of PBP2 and PBP4 estimated at about 370 molecules/cell, it is clear that the periplasmic concentration of CAZ must reach higher values than FEP to exert an effective antibiotic activity.”

4. Second full paragraph of Page 9: Although cephalosporins are generally considered to be porin translocated, that LPS decreases the permeability of the OM to many antibiotics has been well established. This should be acknowledged in this paragraph and it should be made more clear that the open question would be whether cephalosporins are affected.

-> Indeed, it is well-accepted that modifications of the lipid A moiety of LPS are responsible of resistance towards cationic compounds such as polymyxins and antimicrobial peptides. Target genes *arnT* and *eptA* of the PmrAB two-component system catalyze the addition of amino-arabinose and phosphor-ethanolamine residues to the 4' and 1 ends phosphorylated of the disaccharide (Kdo)₂, respectively. To date, several studies have reported gain-of-function mutations *pmrA* in polymyxin-resistant clinical strains of enterobacteria. For β -lactams, the role of LPS is different insofar as LPS biogenesis is essential for the insertion of porins in the form of active trimers in the outer membrane [Arunmanee W, Pathania M, Solovyova AS, Le Brun AP, Ridley H, Baslé A, van den Berg B, Lakey JH. Gram-negative trimeric porins have specific LPS binding sites that are essential for porin biogenesis. Proc Natl Acad Sci U S A. 2016, 113:E5034-43. doi: 10.1073/pnas.1602382113.; Bolla JM, Lazdunski C, Pagès JM. The assembly of the major outer membrane protein OmpF of *Escherichia coli* depends on lipid synthesis. EMBO J. 1988, 7:3595-9. doi: 10.1002/j.1460-2075.1988.tb03237.x.]. Many others studies have reported a genetic link between porin assembly and LPS biogenesis in *E. coli*. This aspect has been reviewed earlier in Vergalli J, Bodrenko IV, Masi M, Moynié L, Acosta-Gutiérrez S, Naismith JH, Davin-Regli A, Ceccarelli M, van den Berg B, Winterhalter M, Pagès JM. Porins and small-molecule translocation across the outer membrane of Gram-negative bacteria. Nat Rev Microbiol. 2020, 18:164-176. doi: 10.1038/s41579-019-0294-2.

In the present work, whole-cell assays were performed in strains derivatives of *E. coli* W3110. Consequently, it is unlikely that the biogenesis of LPS is affected, the only variable between the strains tested being the nature of the porins expressed. The differences observed between the results obtained using whole cells compared to in vitro tests can indeed be explained by the absence of LPS in the latter. This is clarified in the discussion (page 9).

Reviewer #2 (Remarks to the Author):

The emergence of bacterial strains with antibiotic resistance is a pressing problem for society and the mechanisms of antibiotic resistance including methods to overcome this problem are topics of significant current interest. Based on cumulative literature results, the authors concluded that some antibiotic resistance may originate from the lack of outer membrane porins, which prevents certain antibiotic from reaching their intracellular target. Herein, the authors report a comparative analysis of the porin permeability of several antibiotics through the outer membrane protein F and C (OmpF versus OmpC). OmpF and OmpC were therefore expressed in *E. coli* and the cell toxicity of several antibiotics was assessed by the resazurin assay and compared to the uptake as determined by LC/MS-MS. In addition, the permeability was determined by single channel current measurements in planar lipid bilayers and a fluorescent assay in liposomes.

The authors focused on comparing the cephalosporins ceftazidime (CAZ), cefotaxime (CTX), and cefepime (FEP). The intracellular quantification by LC/MS-MS conclusively demonstrated that the intracellular concentration of CAZ is significantly lower in *E. coli* with OmpC than with OmpF. It is a reasonable conclusion that this may indeed cause an antibiotic resistance of OmpF-deficient bacteria containing only OmpC. Ion conductance measurements with planar lipid bilayers also confirmed a lower permeation of CAZ through OmpC than through OmpF; the permeation of CTX was also reduced through OmpC compared to OmpF, but to a lesser extent, whereas FEP permeation could not be assessed, because the molecule is uncharged. The results from the fluorescent liposome assay indicate an overall increased permeation of FEP compared to CAZ.

Overall, the results suggest that the initial hypothesis, namely that the resistance of some bacteria towards ceftazidime (CAZ) may originate from the lower permeation rates through OmpC pores, holds true. This reviewer, however, still wonders how CTX can show a similarly strong antibiotic activity on OmpF- and OmpC-containing bacteria, when the permeation rates are also different?

-> We thank the referee for this comment. Indeed, whole-cell assays show that CTX has an antibiotic activity similar to that of FEP, whether the bacteria express a type F or C porin. Conversely, CTX shows a low permeation rate through both OmpF and OmpC using zero-current assays (page 8). This behavior can be explained by the fact CTX has a much higher binding affinity to the critical PBPs (i. e. PBP1a, PBP1b, PBP2 and PBP3), or a much lower IC50, as compared to both FEP and CAZ [Rahme C, Butterfield JM, Nicasio AM, Lodise TP. Dual beta-lactam therapy for serious Gram-negative infections: is it time to revisit? *Diagn Microbiol Infect Dis.* 2014, 80:239-59. doi: 10.1016/j.diagmicrobio.2014.07.007.].

Another question is, why the authors did not include liposome assay results with CTX?

-> Using reversal potential measurements, we obtained quantitative translocation numbers for two charged compounds: CAZ & CTX. These are comparable. Neutral molecules cannot be quantified in this way so we searched for another method to characterize FEP. So, we added the liposome study. The competitive study using the pair of reporters encapsulated in CB8/MDAP liposomes only allows a semi-quantitative analysis. Obviously, we can qualitatively reveal rapid translocation. On one hand, the underlying difficulty is controlling the amount of reporter pairs and the amount of functional porins across different batches. The different affinities of the three cephalosporins for the reporter pair are observed in the different plateau levels and agree well with their different affinity for CB8/MDAP. On the other hand, recording the reversal potential only allowed for charged compounds (CAZ & CTX) to obtain quantitative numbers. These are comparable. However, in order to classify FEP, we added the liposome study.

Figure S7 has been changed to show binding of CTX to the CB8/MDAP reporter pair.

Figure S8 (see below) has been added to show comparison of CAZ, FEP and CTX translocation across OmpF and OmpC using FARMA.

Previous Figures S8 and S9 are now S9 and S10.

All changes have been highlighted in the text.

Figure S8. Changes in the fractional MDAP emission ($\lambda_{ex}=339\text{ nm}$, $\lambda_{em}=422\text{ nm}$) of CB8/MDAP-loaded liposomes upon addition of a) OmpF or b) OmpC (45 nM, at 60 s), antibiotic (26 μM , at 120 s) and tryptophan amide (25 μM , at 600s) for calibration.

Reviewer #3 (Remarks to the Author):

This manuscript describes a detailed study of porin-based transport of a variety of clinical stage cephalosporin analogs into Gram-negative bacteria. Resistance to beta-lactam analogs in Enterobacteriaceae is an important health threat, especially for extended-spectrum carbapenems. This

manuscript takes a variety of genetic and biophysical approaches to quantitating the dependence of several different agents on different bacterial porins. The work is carefully done and is well described. The data is presented with good reproducibility and statistical treatment. The authors determine that different beta-lactam chemotypes utilize porins differentially and suggest that potential resistance mechanism could arise from deletion or mutation of specific porins. While the work is of high quality and there is nice use of an innovative technique to directly monitor the transport of compounds across membranes, the manuscript is very descriptive in nature. There is little in the way of detailed analysis linking chemical descriptors to specific transporters, and it seems primarily that these techniques would primarily be useful in a retrospective manner. The importance of the work would be greatly enhanced if testable models could be generated and used to predict porin specificity of new agents or how to enhance porin transport by chemical modification of an existing agent.

-> We agree with this comment. Indeed, this was one of the goals of the IMI project Translocation to find a new Lipinski rule of five or a particular scaffold with good penetration (see the work of the Hergenrother laboratory). Unfortunately, this has not (yet) succeeded. In the meantime, we try to go beyond being purely descriptive but to extract reproducible quantitative data by combining a wide range of techniques.

It is also not clear how important altered porin-transport is to drug resistance, this was often invoked for fluoroquinolones but clinical surveillance shows very little of this compared to gyrase mutations. Perhaps the work could be enhanced by securing a panel of cephalosporin-resistant isolated and determining if alteration to porins is seen in the wild.

-> We thank for the referee for this comment and fully agree. The present work has been initiated following numerous publications, which analyze the expression of porins in infected patients under antibiotic treatment [Int J Antimicrob Agents. doi:10.1016/j.ijantimicag.2012.10.010; Antimicrob Agents Chemother. doi: 10.1128/AAC.01585-15; Antimicrob Agents Chemother. doi: 10.1128/AAC.01607-09.] and a recent review on this domain [Nat Rev Microbiol. doi: 10.1038/s41579-019-0294-2.]. All these publications and several others report a “sequential evolution in porin production”: first, both OmpF and OmpC type porins are produced, then bacteria exhibit a loss of OmpF over OmpC and finally no porin at all (or porins with functional mutations).

We also plan to correlate porin expression to CAZ and FEP susceptibility in our laboratory collection of clinical strains in relation with the prescribed antibiotic therapy of the patients.

I think this is very nice work to study these transport phenomena but that the manuscript needs to move beyond these initial studies to merit publication in Communications Biology.

-> The thank the referee and hope the revised manuscript will be well considered for publication in Communications Biology.

Reviewer #2 (Remarks to the Author):

My comments have been addressed. The authors included additional result to support their arguments revised the manuscript accordingly.

Reviewer #3 (Remarks to the Author):

I think the authors have done a reasonable job of addressing the comments they could form prior reviews. They have increases the technical quality of the manuscript and provided some needed details. I still have some concerns that the the overall impact of the work may be limited but on balance this is a nice body of work and merits publication in Communications Biology